# SCOBY Cellulose Modified with Apple Powder—Biomaterial with Functional Characteristics

**DOI:** 10.3390/ijms24021005

**Published:** 2023-01-05

**Authors:** Malgorzata Anita Bryszewska, Erfan Tabandeh, Jakub Jędrasik, Maja Czarnecka, Julia Dzierżanowska, Karolina Ludwicka

**Affiliations:** 1Faculty of Biotechnology and Food Sciences, Institute of Natural Products and Cosmetics, Lodz University of Technology, 2/22 Stefanowskiego St., 90-537 Lodz, Poland; 2International Faculty of Engineering, Lodz University of Technology, 36 Zwirki St., 90-539 Lodz, Poland; 3Institute of Molecular and Industrial Biotechnology, Lodz University of Technology, 2/22 Stefanowskiego St., 90-537 Lodz, Poland

**Keywords:** SCOBY, bacterial cellulose, vegan leather, biomaterials, apple powder, waste materials

## Abstract

The need for new non-animal and non-petroleum-based materials is strongly emphasized in the sustainable and green economy. Waste materials have proven a valuable resource in this regard. In fact, there have been quite a large number of goods obtained from wastes called “Vegan leather” that have gained the clothing market’s attention in recent years. In practice, they are mostly composites of waste materials like cactus, pineapples, or, eventually, apples with polymers like polyurethane or polyvinyl chloride. The article presents the results of work aimed at obtaining a material based entirely on natural, biodegradable raw materials. Bacterial cellulose produced as a byproduct of the fermentation carried out by SCOBY was modified with glycerol and then altered by the entrapment of apple powder. The effect of introducing apple powder into the SCOBY culture media on the mechanical properties of the obtained bacterial cellulose was also evaluated The resulting material acquired new mechanical characteristics that are advantageous in terms of strength. Microscopic observation of the apple powder layer showed that the coverage was uniform. Different amounts of apple powder were used to cover the cellulose surface from 10 to 60%, and it was found that the variant with 40% of this powder was the most favorable in terms of mechanical strength. Also, the application of the created material as a card folder showed that it is durable in use and retains its functional characteristics for at least 1 month. The mechanical properties of modified bacterial cellulose were favorably affected by the entrapment of apple powder on its surface, and as a result, a novel material with functional characteristics was obtained.

## 1. Introduction

Current sustainable and green economy trends also include finding innovative uses for waste and creating new materials from it that are as kind to the eye as they are to the environment. A sustainable biomaterial derived from natural sources such as plants and microorganisms is not only environmentally friendly and biodegradable but it also helps to reduce textile waste [1]. “Vegan leather” continues to gather attention as a next-generation alternative to natural leather, being environmentally friendly and produced without the use of any animal components. This trend is already strong, and there has been a wide range of materials prepared using natural latex, plant fibers (for example: from apples, pineapples, and cacti) and cell-cultured skins (mycelium). Agricultural waste is increasingly and widely used to produce more environmentally friendly materials—but the adjective ‘more’ is key, since they are not exactly plastic-free. Most of the natural or synthetic composites are combined with polyurethane (PU) or polyvinyl chloride (PVC) or other elastomeric synthetic polymers having a long-chain linear molecule arrangement [2]. The harmful environmental effects of PU and PVC processing are a major problem [3]. PVC production has negative impacts on the environment, with a high contribution to the human toxicity potential (emissions of dioxins, sulfur dioxide, nitrogen oxides [NOx] and global warming potential) [4]. Their contents of dangerous chemical additives like chlorine, phthalates, and lead are also a downside [5,6]. Due to environmental pollution and non-biodegradable components, “Vegan leather” only partially satisfies expectations.

Our interest in sustainable materials and the desire to find a material that could replace or at least reduce the use of PVC and PU in vegan leather led us to turn our attention to bacterial cellulose (BC). This polysaccharide is synthesized as an extracellular biopolymer by several bacteria belonging to the genera of *Acetobacter, Achromobacter, Komagataeibacter, Agrobacterium, Bacillus, Azotobacter, Sarcinia, Lactobacillus*, and *Gluconacetobacter*. The structure of the BC is formed by the inter and intramolecular bonds of hydroxyl groups with supramolecular interactions, resulting in the packing of the cellulose fbrils in a fbrillar and semi-crystalline form [7]. BC is characterized by its unique purity and ultrafine structure. The microfibers of BC are typically about 100 times smaller than those found in plant cellulose. The characteristics of its mechanical, morphological, and biological properties have shown that cellulose synthesized by bacteria has a molecular structure identical to that of plant cellulose, but that it also has higher degrees of purity, polymerization, and crystallinity and therefore greater tensile strength, water absorption, water retention capacity, and biological adaptability, offering both biocompatibility and biodegradability [8,9]. Unlike plant cellulose, BC has no lignin or hemicellulose, which is why it does not need to undergo delignification and detoxification and can be processed with ease via mild alkali treatment [10]. Moreover, it has a large surface area, which allows for the adsorption of different liquids and the formation of strong bonds with other biomaterials, polymers, antimicrobial substances, enzymes, and nanoparticles [7,11]. BC composites are obtained through various modifications either introducing a modifier prior to biosynthesis (in situ) or after biosynthesis (ex situ) [12]. The change in the material properties caused by introducing a modifier in situ is mostly driven by a change in the intrinsic biophysical properties. In turn, the incorporation of molecules into existing cellulose fibers results in a change in their properties by modifying the porosity and/or BC crystallinity [13,14,15]. The physical entrapment method can be employed to capture proteins directly inside BC without changing the BC’s shape. This method not only improves the durability of BC, it also minimizes the modification process. The biopolymer can be impregnated with reinforcement materials such as polymers, metals, nanomaterials, and antimicrobials with carboxylation, amidation, acetylation, and other chemical reactions [16]. These unique attributes make BC attractive from a technological, ecological, and financial perspective and have resulted in numerous applications. Examples of use can be found in environmental biotechnology [17], bio-processing [18], textiles [19] and biomaterials [20,21,22], wound dressings [23], implants and scaffolds for the tissue engineering of cartilage, as well as carriers for drug delivery [24,25,26], raw material for food and as food packaging [27], and polymer electrolyte membranes [8].

Bacterial cellulose, BC-derived biomaterials and composites have been extensively studied in the last decades due to possibility of obtaining new structures with remarkable properties. However the majority of studies have used cellulose produced by isolated, well-defined strains with relatively quick and high productivity [28]. In this work, we chose to use BC produced by symbiotic colony of bacteria and yeasts (SCOBY), since this method of production is more cost-effective, does not require the availability of a specific bacterial strain(s), and is accessible even under domestic conditions. The biofilm is naturally formed as a byproduct during the process of juices fermentation or kombucha beverage fermentation, and it is observed as a pellicle at the air–liquid interface. In terms of composition and structure, the biopolymer corresponds to cellulose. The composition of the microbial population in SCOBY may vary according to its origin, geographical location, weather, and the medium used for the fermentation process. However, there are some general similarities. Predominantly, Acetobacter bacterial species, various Saccharomyces, and a number of other types of yeasts are presented [29]. The yeast cells of the inoculum metabolize the sugar in an invertase-mediated reaction, and the hydrolysis of sucrose leads to glucose and fructose production which, through the glycolysis pathway, is converted into ethanol. *Gluconobacter* and *Komagataeibacter* bacteria metabolize glucose to produce gluconic acid, and in addition, they may polymerize glucose residues, thereby forming a BC mat supporting the microbial culture [29]. A disadvantage of choosing SCOBY is the longer time needed to produce a suitable biofilm relative to that needed for its production with, for example, *Komagataeibacter xylinus*.

The application of BC as bio-leather has some drawbacks, however they can be overdrawn by biopolymer modification. In the current research AP was selected as the BC properties modifier. The practical reasons behind our choice were both the availability and the price of it. AP is a waste material generated by the fruit industry that is both readily available and inexpensive. A further factor influencing the choice was the chemical composition of pectin and cellulose. The expectation was that the similarities in the structures would make it possible to form bonds between the biopolymers, and that as a result, a durable material could be formed.

## 2. Results and Discussion

### 2.1. The Morphology of Materials

The digital and SEM images of the BCN and AppBCs are presented in Figure 1. The yellowish color of the native biopolymer without any treatment is associated with the residues of tea extract. Pre-treatment by bleaching and swelling followed by modification with glycerol as a plasticizer results in a white and plastic material (mG). It feels slippery to the touch and gives the impression of being slightly greasy. Entrapping of apple powder gives the material a characteristic color of dried apples (from mGApp10 to mGApp60 as well as G_mGApp30 and G_mGApp50). The increasing concentration of AP in the modifying mixture is reflected in the final product, which is easily noticeable by the deepening color of the materials as this concentration increases. This modification also makes the AppBCs more pleasant to the touch—the slippery sensation disappears. When comparing AppBCs obtained from fermentation with and without the presence of AP, it can be observed that the color of the former is paler.

More-detailed structural and morphological characteristics of mG and AppBCs can be observed on the images generated by scanning electron microscope (SEM) presented in Figure 2. The SEM images showed fine cellulose fibers that form a porous three-dimensional network structure. The compact cellulose cross-linked net shows a lattice structure consisting of fibers with no apparent difference in size. The general assembly of fibrils consists of organized regions, but fragments with random arrangements can also be observed.

The AppBCs exhibited different morphologies, and the surface of those materials was smoother than that of mG (Figure 2A,B vs. C–F). The fiber structure was not seen so well as it was in the case of the sample without surface modification. The trapping process produced a layer on the cellulose surface, as shown in Figure 2C–F. The coating is uniform over the entire surface and also fills in the cavities, which is clearly visible in the images where the edges are shown (Figure 2D,F). In SEM images of mGApp50 (Figure 2F) the surface seems to be smother than mGApp10, suggesting that the BC surface is coated with a denser layer. Entrapping appears to be uniform across the entire surface, resulting in a smooth tactile sensation. Such an effect was not observed in studies in which protein isolates were used for entrapment [15]. The formed soy protein isolate agglomerated on the surface and inside the bacterial cellulose.

### 2.2. The Structural Analysis (FTIR)

Observations of the chemical composition and specific structural and conformational differences between the BC samples and their modified variants were performed in the IR spectra. The FT-IR spectra of native BC were compared with those modified by entrapping apple powder.

In the BCN spectrum, the absorption bands are observed in two wave number regions: 3600–2800 cm^−1^ and 1700–600 cm^−1^ (Figure 3A). A strong broad band can be observed in the wave number range of 3600–3000 cm^−1^, which is assigned to different O–H stretching vibration polysaccharides. In this broad band there is a distinguishable peak at 3342 cm^−1^ that is described in the literature as corresponding to the strength of the intramolecular 3OH⋯O5 intramolecular hydrogen bond [30], and it can be observed in the spectra of BCN, BC, and mG, but is not detected in mGApp50. In the absorption band range of 2880–2930 cm^−1^, there is a peak is assigned to the C–H bending bonds. Also, characteristic peaks of cellulose at 1420 and 1310 cm^−1^ represent the symmetric stretching and out-of-plane wagging of the CH_2_ groups, respectively [30]. The absorption bands at 1367 cm^−1^ belong to vibrations of –CH. Characteristic for carbohydrates’ IR signals in the region of 1160–1033 cm^−1^ corresponding to the C–O–C antisymmetric bridge stretching of β-1,4-glycosidic bond, C–C, and C–O stretching vibrations, C–O–H bending and C–O–C pyranose ring skeletal vibration can also be observed. The changes in the structure of the native cellulose caused by pretreatment can be seen in an intense band with the pick at 1585 cm^−1^ wavenumbers. The pick corresponds to carboxylate antisymmetric stretching bands (COO^−^). Apparently, treatment with hydrogen peroxide caused oxidation, and COO^−^ groups were formed. Further modification of BC by glycerol affected this band, as can be seen in the spectrum of mG where this band has a much lower intensity, and a new peak can be observed at slightly higher wavenumbers. The main bands in the mG spectrum are also present in the mGApp50. However, differences in the absorption band intensities in spectra-modified cellulose samples are observed. Comparing the FTIR spectra before and after modification of the bacterial cellulose with apple powder, the band at the wavenumber 3340 cm^−1^ corresponding to O-H bond stretching was more curved, indicating changes in the intensity value. That observation is in accordance with the literature and could be related to the interlinkage of pectin and BC fibers through H bonding [31]. Concurrently, there was an intensity increase of the C-H bending bond in the range of 2990–2800 cm^−1^ wavenumbers showing intramolecular bonds. The changes observed in the BC spectra suggest that both modifiers were incorporated in the structure of BC as a result of chemical interactions between cellulose and glycerol and as well between pectin.

Comparison of the spectra of the materials obtained after fermentation with AP (G_mG, G_mGApp30, and G_mGApp50) reveals subtle changes, such as increased intensity of the 1650 cm^−1^ peak when G_mG was modified with AP (Figure 3B). Such an observation seems to be expected; in fact, AP was present in all of these materials, and the same interactions or bonds were formed. Moreover, these spectra are similar to the one recorded for mGApp50 (Figure 3A). As was observed before, the pick at wavenumber 1585 cm^−1^, assigned to carboxylate antisymmetric stretching bands (COO^−^), has lower intensity in the spectra recorded for the materials modified with apple powder, providing that the group is involved in AP bonding in the analyzed samples.

### 2.3. Tensile Strength

Materials (Gm, m_GM and AppBGs) were tested for tensile strength (MPa), elongation at break (%), Young’s modulus, and load at maximum force. BCN was not subjected to this test, because it did not have practical value, among other reasons, and because it could contain residual microorganisms and a culture solution making it yellow over time. Similarly, BC was not taken into consideration in the presentation of results, since the material differed greatly from the others. The lack of plasticizer made it more prone to tearing and mechanical damage than its glycerol-modified counterparts, though its mechanical parameters were comparable to glycerol-modified materials (data not presented in the article). Additionally, BC was less attractive than glycerol-modified samples from both the visual and touch point of view. As presented in Figure 4, the introduction of apple powder into the SCOBY growing media worsened the mechanical parameters of the glycerol-modified material (mG and G_mG), and this difference was statistically significant with the exception of the Young’s modulus, which remained similar for both types of materials. This observation is consistent with the literature, presenting properties of biocomposites obtained by adding pectin and other polysaccharides to the *G. xylinus* media [32,33] that addition of pectin to the *G. xylinus* growth media resulted in biocomposites with lower mechanical strength than native bacterial cellulose [31]. The decreased tensile and tear strength of the BCP biocomposites was due to the increasing amorphous region in the BCP structure, as proven by the decreasing crystallinity observed under XRD analysis and FTIR spectra.

As well as for materials subjected to modification by trapping apple powder, mechanical parameters were affected by the variant of culture (mGApp30 vs. G_mGApp30 and mGApp50 vs. G_mGApp50) and were higher for the BC grown in media without App; however, the differences were not statistically significant.

A comparison of the mechanical parameters of the materials obtained by trapping apple powder at concentrations ranging from 10 to 60% showed that the use of 40% AP in the modifying mixture was the most favorable. All four parameters were higher for the mGApp40 material, and in the case of tensile strength and Young’s Modulus, this difference was statistically significant. Further increases in App concentration resulted in a decline in the mechanical performance of these materials.

### 2.4. Water Resistance

One of the most crucial properties of biomaterials, one which significantly determines their use, is their resistance to moisture. In order to determine this property for materials, such parameters as swelling capacity, water absorbency time, and water contact angle are determined.

The capacity of swelling was expressed as the weight change of the samples during water soaking was measured after 5, 10, 30, and 120 min, and is represented in Figure 5. All materials absorbed water, but the extent of mass increase widely varied. After 2 h of soaking, the weight increase ranged from 131 to 1168% (for BC and G_mGApp50, respectively), which means it was nearly 10 times higher for the material with the strongest water absorption properties. The weakest swelling properties were shown by BC, additionally, for this material, the water absorption plateau was reached after the first 10 min. The only material for which swelling did not appear to reach saturation, at the time of observation, was mG. The presence of AP in the materials increased the swelling level by up to 493% for samples grown in media without AP (mGApp30) (Figure 5A) and up to 1168% for materials grown in the presence of AP (G_mGApp50) (Figure 5B). A correlation can be observed for the latest materials between the amount of AP used for modification and the resultant swelling. The final mass increase of G_mGApp50 was greater than that of G_mGApp30, and the difference was 268%. Additionally, in samples grown without AP, increasing the AP concentration for modification had an effect on these properties, but the maximum increase in weight was only 493% (for mGApp30) [34]. The higher water absorption properties of AppBCs in comparison with BC are attributed to the fact that the modifier present in those materials is hydrophilic. Apple fiber exhibits strong swelling properties; Negi et al. determined swelling power for apple pomace powder as 1265.20% [34]. The modification with AP in situ and ex situ (G_mGApp30, G_mGApp50) resulted in a higher swelling capacity of the material as compared to the analogous one, but modified only ex situ (mGApp30, mGApp50). AP modification in situ and ex situ (G_mGApp30, G_mGApp50) resulted in a higher swelling capacity of the material in comparison with analogical ones but modified only ex situ (mGApp30, mGApp50). The modifier added before the biosynthesis of cellulose fibers probably had an impact on the structure of biopolymer, changing its porosity and/or crystallinity [35]. Water absorption was determined as the time required to absorb 0.1 cm^3^ of water. It was observed that BC modification with glycerol reduced the time required for water absorption in the materials’ surface (mG and G_mG), which is consistent with literature data [36]. This can be attributed to the fact that modifier is hydrophilic in nature. The presence of the –OH group in its structure assisted in absorbency phenomenon. An opposite absorption behavior was presented by AppBCs. For these materials, the absorption time was longer by up to 20 min as compared to BC (mGApp50). The absorption of water from the surface of the mGApp60 was impaired by cracks and leakage of liquid between them, and it must be taken into consideration that the reported time corresponds to absorption from a larger surface area.

The results of AppBCs’ water contact angle measurements indicate a reduction in the hydrophilicity of the materials over BC, but it remains at a level that is not satisfactory for leather-like materials and suggests the need for further modifications in order to give to the materials wider potential for application.

The solubility of the materials was determined after soaking in distilled water for 2 h. The results are summarized in Table 1. The solubility index ranged from 4.52% to 6.14% (BC and G_mG, respectively), meaning that the maximum range of change was only 1.62%. No statistically significant differences were observed between samples.

### 2.5. Preliminary Studies on Possible Applications

#### 2.5.1. Expanding the Color Spectrum of Materials

When developing this alternative vegan leather material, it was obvious to us that if it is to serve as a replacement for materials already on the market, it must be easy to dye. Leather products typically need to come in a wide variety of colors in order to meet different customer requirements. The materials obtained, AppBCs, were in tones of brown with lighter and darker pigmentation, which produces an obvious association with natural leather. In order expand the range of colors, the AppBCs samples have therefore been dyed. The result of this treatment is presented in Figure 6. As this simple procedure showed, the color of the material can be easily modified, thus increasing its usefulness and expanding its range of applications.

#### 2.5.2. Prototyping

The primary goal of developing AppBCs was to create an alternative to natural leather and leather-like material. A card folder was prepared as a prototype. Pieces of mG and mGApp50 were joined by gluing (Figure 7). The prototype was actively used for 1 month, and during the use time, it was exposed to friction, scratches, etc. After the testing time, no visible changes or signs of exploitation were observed. The material remained durable and flexible. It was easy to notice that the mGApp50 was considerably smoother.

## 3. Materials and Methods

### 3.1. Reagents

Solutions of hydrogen peroxide (H_2_O_2_ 30.0 ± 1.0%, Chempur, Piekary Śląskie, Poland); sodium hydroxide (pellets Merck Sp. z o.o., Warszawa, Poland); acetic acid (99.5–99.9%, Avantor Performance Materials, Gliwice, Poland); glycerol (Chempur, Piekary Śląskie, Poland); and ethanol (96%, StanLab, Lublin, Poland) were prepared using a procedure-based method. All of the reagents were of analytical grade and used without further purification. Kombucha pellicle (SCOBY) were supplied by “Kameleon Kulinarny”, Joanna Grzybowska Łódź, Poland. Powder of apple fiber (food grade) was purchased from Aura Herbals sp. z o.o. Gdańsk, Poland.

### 3.2. SCOBY Bacterial Cellulose Production and Modification

#### 3.2.1. Bacterial Cellulose Production

Bacterial cellulose (BC) was obtained through the process of bioconversion use for Kombucha. Briefly, an infusion of black tea was prepared with 2 L of boiling water, 200 g sugar, and 5 teabags (Lipton Yellow Label), and it was transferred to a rectangular plastic container. After cooling the mixture to room temperature, the symbiotic culture of bacteria and yeast (SCOBY) was introduced into the media. A variant of the culture was also prepared, in which apple powder (AP) was additionally introduced into the culture medium—the content of AP was 30% or 50% *w*/*w* (mass_AP_/mass_wet BC_).

Fermentation was carried out under static conditions in the darkness for about 3 weeks, until the BC layer reached a thickness of approximately 1–2 cm. Thereafter, the fermentation process was finished, and the cellulose pellicle that formed at the solution surface was gently removed from the growing media.

#### 3.2.2. Bacterial Cellulose Preparation for Modification (Pre-Treatment)

The pre-treatments of the obtained BC included washing, bleaching, and swelling according to the procedure described by Kim et al. [15]. Cleaning had the goal of rinsing out the remaining solution as well as removing microorganisms, and it was achieved by washing BC in 3% NaOH solution for 90 min at 25 °C with shaking at 70 rpm in a water bath (Vibra, AJL Electronic, Krakow, Poland). After neutralizing the BC by washing it under tap water, the pH of the BC-containing solution was adjusted to 3.0 using acetic acid and shaken for 30 minutes at 70 rpm. Then, the BC was brought to neutralization by washing it with tap water. Subsequently, bleaching with 5% H_2_O_2_ solution was carried out for 60 min at 90 °C while shaking in a water bath at 120 rpm. After the removal of hydrogen peroxide solution by washing with tap water for 3 min, the BC was swollen with an 8% NaOH solution for 30 min in an ultrasonic bath (SW12H, SonoSwiss, Ramsen, Switzerland). Then, BC was washed under tap water for 3 min and placed in an acetic acid solution with the pH adjusted to 3.0 for 30 min incubation with shaking at 70 rpm. Finally, the acetic acid solution was disposed of, and the treated BC was washed under tap water until a neutral pH was reached. As prepared, such raw BC material was used as a reference for further tests.

#### 3.2.3. Bacterial Cellulose Modification

A procedure previously described by Kim [15] was adapted for the modification of the BC surface by entrapping apple powder. The modifying mixture was prepared from apple powder (AP) and glycerol. The AP was used in concentrations ranging from 10 to 60% of the wet weight of BC. AP powder was suspended in water, the amount of which depended on the weight of BC and was 10 times its weight. In order to obtain a uniform mixture, homogenization was applied. Next, glycerol (30%) was added, and the mixture was first homogenized for an additional 2 min and then ultrasonicated for 30 min at 25 °C. After pH adjustment to 10.0 (±0.5) using 1 M NaOH, the mixture was incubated at 80 °C in a water bath with shaking at 70 rpm for 20 min. The procedure resulted in activated AP in solution with glycerol. Pre-treated BC was immersed and completely covered with a mixture of activated AP and glycerol and then ultrasonicated for 30 min at 25 °C. Next, the entire mixture was transferred to a water bath (35 °C) and shaken at 80 rpm for 1 h. Finally, materials—NBC, BC and AppBCs (BC modified with AP—with different AP concentrations) were oven-dried under mild conditions (35 °C, 4 days). The variants of the modifications carried out are summarized in Table 2.

### 3.3. Mechanical Properties Analysis

The samples of native bacterial cellulose and its variants obtained by modification with apple powder were examined for their tensile strength at breaking using a universal testing machine (Zwick Roell 1 kN, type Xforce HP, S/N:764916, Germany). The mechanical parameters, characteristics—i.e., Young’s modulus, maximum force (Fmax), stress (σ), and elongation at break (ε)—were determined using the TestXpert II software program (version 3.61, ZwickRoell GmbH & Co.KG, Ulm, Germany, 2015). Stress was equal to the loading force, expressed in Newtons (N), divided by the cross-sectional area, measured as the width × thickness of the sample (m^2^). The strain was calculated as DL/Lo × 100%, where Lo is the initial length and ∆L represents the change to the original length after breakage. The values of Young’s modulus were determined on the basis of the linear stress/strain relationship. Based on the mechanical test results, the compressive and breaking strength of the tested materials were calculated from the following equation:Strength [N/m^2^] = (Fmax/cross section area).

The measurements were performed in at least three replicates for each type of material, in an air-conditioned laboratory, at a temperature of 20 °C. Before tensile strength analysis, the materials were cut into 2.0 cm-wide strips. The experiment was carried out with parameters adjusted to 25 mm/min of clamps’ movement velocity and 15 mm of initial distance between the clamps.

### 3.4. Scanning Electron Microscopy (SEM)

Scanning electron microscopy (SEM) of the BC and modification of it was performed using a Jeol-JCM scanning electron microscope (model JCM-6000 Akishima, Tokyo, Japan 2014). The samples were placed in the sample holder and covered with double-sided carbon electroconductive tape, then covered with a layer of gold (0.5–1 nm thickness). The images were recorded at acceleration potential of 5 kV.

### 3.5. Fourier Transform-Infrared Spectroscopy (FT-IR)

The analyses were performed on native BC and BC modified by glycerol and by entrapping AP. The IR spectra were obtained on a Nicolet 6700 (Thermo-Scientific, Waltham, MA, USA) FT-IR spectrometer, in the 4000–600 cm^−1^ region at 2 cm^−1^ spectral resolution and 64 scans per sample. The background spectra were collected before each sample in order to eliminate the signals of the spectrometer and its environment from the sample spectrum. Spectral data were processed using the software of the spectrophotometer (OMNIC ver. 8.0—Thermo Fisher Scientific Inc., Waltham, MA, USA).

### 3.6. Resistance to Water

The swelling degree (%) of samples was determined by dipping the sample of materials (sample size: 1 cm × 1 cm and mass from 0.06 to 1.30 g) into 10 cm^3^ of distilled water for 5, 10, 30, and 120 min. The samples’ weights were measured before the test and after set intervals. The swelling percentage (%) was expressed as (W_S_ − W_0_)/W_0_) × 100%, where W_0_ was the initial weight and W_S_ was the weight after swelling at specified time periods.

The water absorbency of the materials was measured as the time needed to absorb 0.1 cm^3^ of distilled water dropped vertically on the sample surface (sample size: 1.5 cm × 4 cm) from a specified height of 1 cm in a room temperature of 25 °C.

The water contact angle was measured by placing a water drop 10 µL in volume on the materials’ surface using a micropipette. The sample with a deposited water drop was illuminated from one side, and an image of the drop was taken using a mobile phone camera (Oppo, Reno 3) with the application of Magnifying Glass (version 1.3.4. by App2U). The contact angle was calculated using ImageJ software [37].

The solubility index (%) was determined after 2 h of soaking. The solutions after the swelling test were collected in weighed vials and dried at 80 °C for 16 h before being weighed again. The change of vials’ mass (W_solub_) was used for calculation of the solubility index (%) expressed as: (W_solub_/W_0_) × 100%.

### 3.7. Dyeing

A commercial preparation dedicated for home use to dye cotton, linen, silk, etc., was used: the ginger dye was sourced from Argus (Zakliki z Mydlnik 16D, Krakow, Poland), and the black dye was from Dr Granosik, Wytwornia Chemiczna (Zgierz, Poland). The contents of the sachet (15 g of ginger dye) were dissolved in 2 L of hot water, and another 2 L of water and a handful of table salt were added. AppBCs samples were immersed in the prepared solution and heated to a boil with constant stirring. Black dye (10 g) was dissolved in 1 L of warm water, and AppBCs samples were inserted. The bath was heated to a boil with constant stirring and maintained at boiling for 30 min. After that first part of heating, a handful of table salt was added the bath and heated for another 30 min. In both cases, AppBCs were left in the solution to cool down and then rinsed out with tap water several times until the dye stopped leaching out. The materials were oven-dried under mild conditions (35 °C, 4 days).

### 3.8. Statistical Analysis

Statistical analysis of the data was performed using R computational language [38], R Core Team (2022). R: a language and environment for statistical computing. R Foundation for Statistical Computing, Vienna, Austria. URL https://www.R-project.org/ (accessed on 27 October 2022). For post-hoc tests, the package Agricolae: Statistical Procedures for Agricultural Research (version 1.3-5), was used [39]. The data were analyzed using a one-way ANOVA (*p* < 0.05) followed by multiple comparisons using a Tukey’s test, with significance set at *p* < 0.05.

## 4. Conclusions

The parameters of the material obtained as a result of covering the previously modified bacterial cellulose with a plasticizer such as glycerol undergo a significant and beneficial change as a result of trapping apple powder. The apple powder on the surface of the cellulose gives it a uniform coating over the entire surface in the form of a solid, homogeneous and uncracked layer. The material takes on a color in brown tones depending on the amount of AP used in the modification process. However, this color can easily be changed by dyeing, because the material accepts garment dyes, thus expanding its application potential. Another unique feature of the materials is their smell: a delicate natural aroma of dried apple. However, the material is of limited use, as it is not waterproof, and would therefore require further modifications to limit its hydrophilicity.

## Figures and Tables

**Figure 1 ijms-24-01005-f001:**
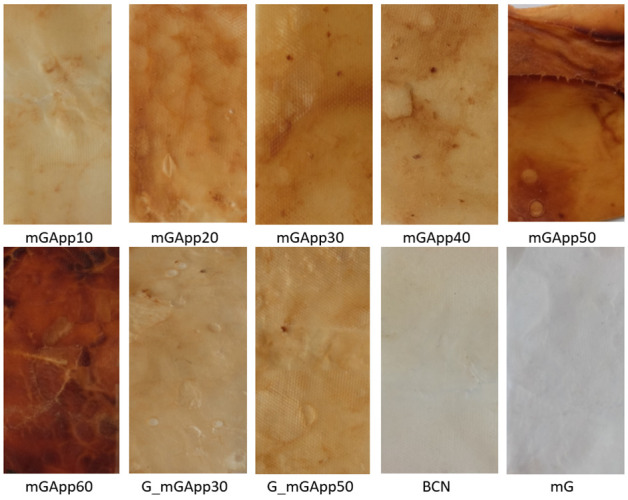
Morphology of the native bacterial cellulose (BCN); bacterial cellulose pre-treated by bleaching and swelling followed by modification with glycerol (mG); BC pre-treated and then modified by glycerol followed by the entrapment of apple powder at concentrations ranging from 10 to 60% (mGApp10, mGApp20, mGApp30, mGApp40, mGApp50, and mGApp60); and BC produced in media with AP, pre-treated, and subsequently modified with glycerol and AP at a concentration of 30% (G_mGApp30) or 50% (G_mGApp50).

**Figure 2 ijms-24-01005-f002:**
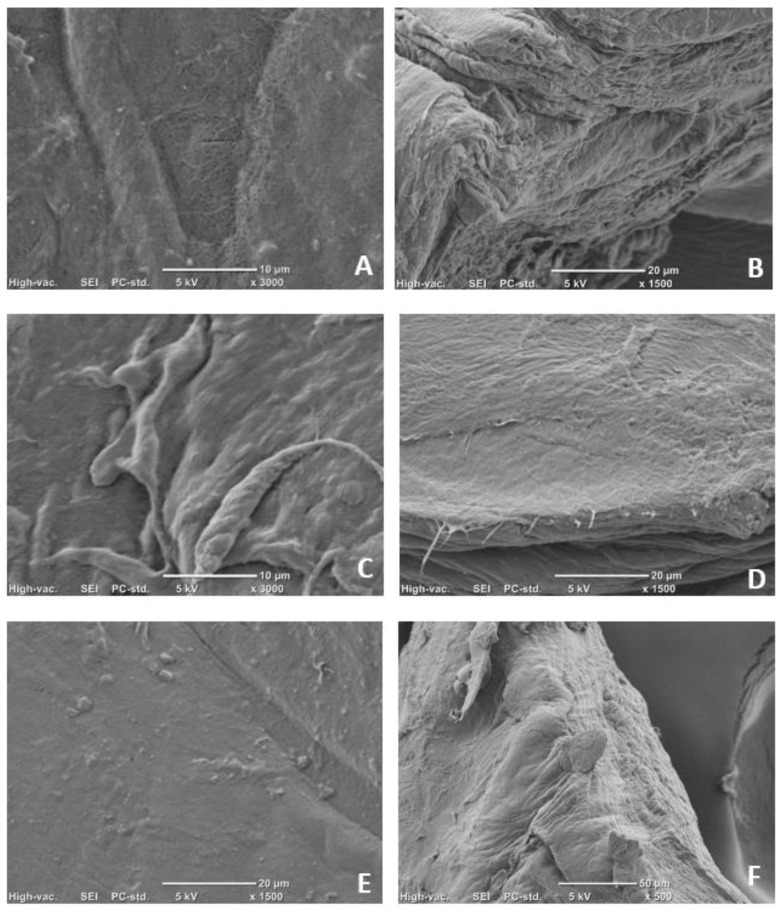
SEM images of materials. Image (**A**,**B**) bacterial cellulose pre-treated and modified with glycerol; (**C**–**F**) bacterial cellulose pre-treated and modified with glycerol and then entrapping apple powder at concentrations of 10% (**C**,**D**) and 50% (**E**,**F**). Images (**A**,**C**,**E**) show the view of the surface, whereas (**B**,**D**,**F**) show edges.

**Figure 3 ijms-24-01005-f003:**
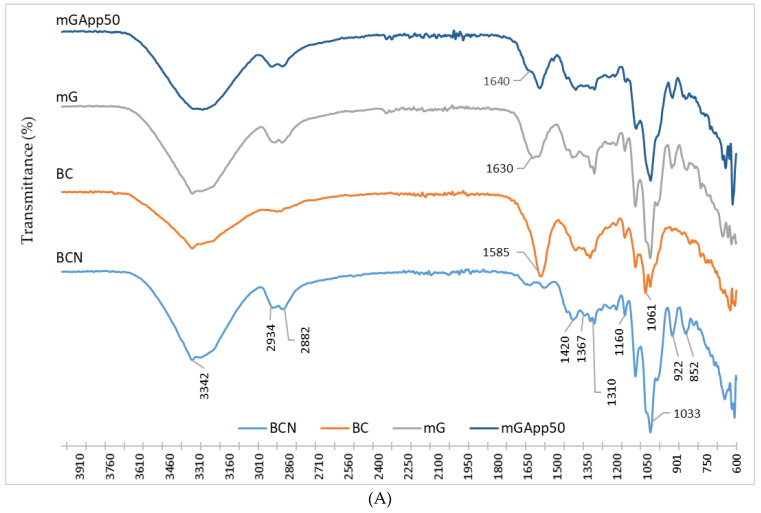
FTIR spectra of bacterial cellulose and the modified materials. The FTIR spectra of bacterial cellulose (BC) are compared with mG (BC pre-treated and modified with glycerol) and mGApp50 (BC pre-treated then modified with glycerol and AP at a concentration of 50%) in (**A**), whereas the spectra of BC grown in the media with AP (G_mG—BC produced in the media with AP, pre-treated and modified with glycerol; G_mGApp30 and G_mGApp50—BC produced in the media with AP, pre-treated followed by modification with glycerol and AP at concentrations indicated by the number) are compared in (**B**).

**Figure 4 ijms-24-01005-f004:**
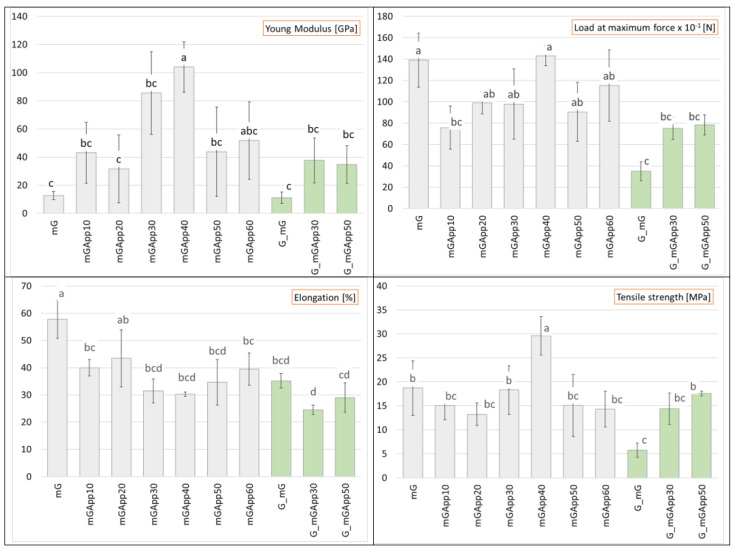
Mechanical strength of the materials: bacterial cellulose modified with glycerol (mG and G_mG); bacterial cellulose pre-treated and modified with glycerol and by entrapping apple powder (mGApp10; mGApp20; mGApp30; mGApp40; mGApp50; mGApp60; G_mGApp30; and G_mGApp50). Bacterial cellulose produced in media without (presented in grey) or with apple powder (presented in green). Different letters represent significant differences.

**Figure 5 ijms-24-01005-f005:**
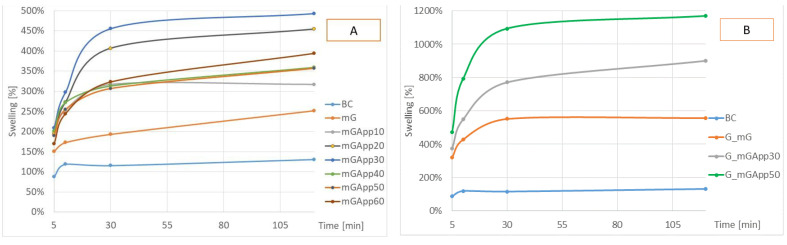
The swelling capacities of materials: bacterial cellulose (BC); bacterial cellulose pre-treated and modified with glycerol (mG and G_mG); bacterial cellulose pre-treated and modified with glycerol and by entrapping apple powder (mGApp10; mGApp20; mGApp30; mGApp40; mGApp50; mGApp60; G_mGApp30; and G_mGApp50). (**A**) Bacterial cellulose produced in media without apple powder (mGApp10; mGApp20; mGApp30; mGApp40; mGApp50; and mGApp60); (**B**) bacterial cellulose produced in media with apple powder (G_mGApp30; G_mGApp50). All data are the average of 3 results.

**Figure 6 ijms-24-01005-f006:**
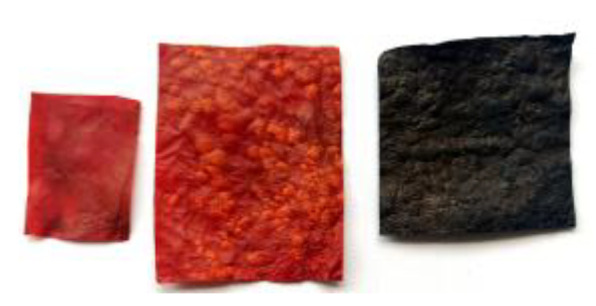
The pieces of material (mGApp50) after the dying process.

**Figure 7 ijms-24-01005-f007:**
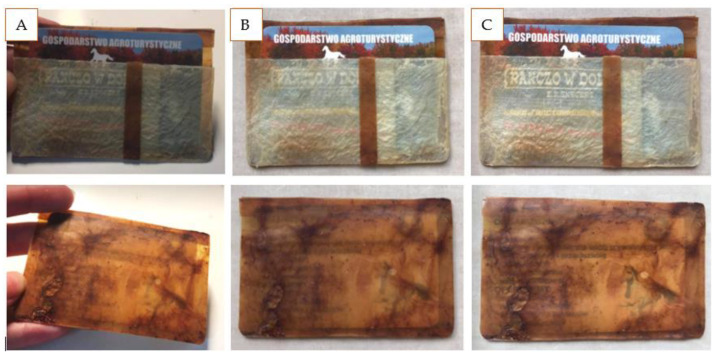
The card holder prepared as a prototype (**A**) and the changes in the material after 1 week (**B**) and 1 month (**C**).

**Table 1 ijms-24-01005-t001:** Water absorbency behavior of the materials. The results are presented as an average (n = 3) ± standard deviation.

	Water Absorbency in Time	Water Contact Angle	Solubility Index
Sample code	[min]	[°]	[%]
BC	40 ± 12	wetting	4.52 ± 2.72
mG	23 ± 6	wetting	5.30 ± 0.98
mGApp10	53 ± 5	121	5.64 ± 0.11
mGApp20	42 ± 4	121	5.58 ± 0.22
mGApp30	43 ± 2	120	5.62 ± 0.13
mGApp40	49 ± 4	122	5.63 ± 0.06
mGApp50	66 ± 2	94	5.12 ± 0.16
mGApp60	28 ± 9	93	4.97 ± 0.31
G_mG	8 ± 1	wetting	6.14 ± 0.09
G_mGApp30	37 ± 3	104	5.13 ± 0.14
G_mGApp50	47 ± 5	96	5.12 ± 0.16

**Table 2 ijms-24-01005-t002:** The conditions used to produce bacterial cellulose and its subsequent modification. Samples codes: BCN—bacterial cellulose native; BC—bacterial cellulose pre-treated; mG—BC pre-treated and modified with glycerol; G_BC—BC produced in the media with AP; G_mG—BC produced in the media with AP, pre-treated and modified with glycerol; mGApp10, mGApp20, mGApp30, mGApp40, mGApp50, and mGApp60—BC pre-treated then modified with glycerol and AP at a concentration indicated by the number; G_mGApp30 and G_mGApp50—BC produced in the media with AP, pre-treated followed by modification with glycerol and AP at a concentration indicated by the number.

Sample Code	Growth Media		Modification		
	Basal Composition	Basal Composition with Apple Powder	Pre-Treatment	Glycerol	Apple Powder [%]
BCN	+				
BC	+		+		
mG	+		+	+	
mGApp10	+		+	+	10
mGApp20	+		+	+	20
mGApp30	+		+	+	30
mGApp40	+		+	+	40
mGApp50	+		+	+	50
mGApp60	+		+	+	60
G_BC		+	+		
G_mG		+	+	+	
G_mGApp30		+	+	+	30
G_mGApp50		+	+	+	50

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
