# Peer review of "SCOBY Cellulose Modified with Apple Powder—Biomaterial with Functional Characteristics"

_ijms, 2023, doi:10.3390/ijms24021005_

Round 1
Reviewer 1 Report
The work titled “SCOBY cellulose modified with apple powder - biomaterial with functional characteristics” requires some correction and supplementation before potential publication.
Is the modification of SCOBY based on adsorption of the apple+glycerol on the cellulose surface without chemical bonding? If yes, I can not see the aim of this study an application of material as vegan leather due to week mechanical properties and low barrier properties and hydrophilicity. I recommend finding other potential application of the modified films.
Bacterial strains names should be in italics.
What
Page 2 line 62 use full abbreviation of the vitamins
Page 3 line 113. This results/feeling indicate a perspiration of plasticizer from polymeric matrix.
Page 6 line 174: remove “at”,
Line 178: add unit to “1420 and 1310”
line 206-208: correct the style/grammar
Fig. 4: : Anyway, BC should be presented as the reference
Page 8 line 268: what was the purity of glycerol and ethanol?
If the aim of the study is applying the material as leather, some investigation of barrier properties or character of the surface should be take into consideration, ie.: surface contact angle, swelling and dissolution degree, moisture absorption.
Author Response
Is the modification of SCOBY based on adsorption of the apple+glycerol on the cellulose surface without chemical bonding? If yes, I can not see the aim of this study an application of material as vegan leather due to week mechanical properties and low barrier properties and hydrophilicity. I recommend finding other potential application of the modified films.
>>> Based on FTIR analysis, we believe that there were changes in the functional groups that resulted from the interaction of cellulose with both glycerol and apple powder. Based on these changes, we assume that the binding is not just of a physical, surface coating of cellulose, but is of a chemical nature. In the pre-treatment procedure, H2O2 was used, which caused the CO- group to oxidize and the COO- group to be formed (we observe a new band). It then changed after the introduction of glycerol, due to the binding of this polyols. After the introduction of AP, changes are observed in the area showing hydrogen interactions, suggesting the formation of bonds of the same nature, but with a new molecular chain.
Bacterial strains names should be in italics.
>>> Notation were corrected. To explain this error, we can say that it was the automatic formatting intervention that we didn't notice.
What
Page 2 line 62 use full abbreviation of the vitamins
>>> The paragraph has been rewritten
Page 3 line 113. This results/feeling indicate a perspiration of plasticizer from polymeric matrix.
>>> We agree with this comment and are aware that glycerol was used in excess for this one-step modification. Based on the work of Cielecka et al. (doi:10.1007/S10570-019-02501-1/FIGURES/7.) we can assume that a glycerol concentration of 2.5% v/v to 10% would be sufficient for this one-step modification. In our work, however, this was a preliminary step, and as it turned out later, the high concentration was beneficial for subsequent modifications.
Page 6 line 174: remove “at”,
>>> it has been corrected
Line 178: add unit to “1420 and 1310”
>>> it has been corrected
line 206-208: correct the style/grammar
>>> The sentence has been changed
Fig. 4: : Anyway, BC should be presented as the reference
>>> Additional explanation has been included to the manuscript. In above we can explain we can explain that to produce AppBCs, we used glycerol and AP as modifiers. In the comparison, we presented only those results that allow us to evaluate the effect of these two modifiers on the strength AppBCs. Therefore, the materials obtained by treating BC with glycerol were considered as reference samples. The effect of glycerol on the properties of bacterial cellulose has been studied and is well described in the literature as for example:Cielecka, I.; Szustak, M.; Kalinowska, H.; Gendaszewska-Darmach, E.; Ryngajłło, M.; Maniukiewicz, W.; Bielecki, S. Glycerol-Plasticized Bacterial Nanocellulose-Based Composites with Enhanced Flexibility and Liquid Sorption Capacity. Cellulose 2019, 26, 5409–5426, doi:10.1007/S10570-019-02501-1/FIGURES/7.
Page 8 line 268: what was the purity of glycerol and ethanol?
>>> The description of reagents has been expanded
If the aim of the study is applying the material as leather, some investigation of barrier properties or character of the surface should be take into consideration, ie.: surface contact angle, swelling and dissolution degree, moisture absorption.
>>> The advised measurements were carried out and the results were included in the manuscript.
Reviewer 2 Report
In this study, Bacterial cellulose as a by-product of fermentation carried out by SCOBY was modified with glycerol and then altered by entrapping of apple powder. The effect of the presence of apple powder in the SCOBY culture media on the mechanical properties of the obtained bacterial cellulose has been evaluated. The topic of the study is interesting, but the manuscript requires major revision before publication.
- - The introduction section is too long and only covers some requirements which should be presented in this section. The introduction usually requires a short review of the literature about the research topic. Then it is best constructed as a descriptive funnel, starting with broad topics and slowly focusing on the current work. One or two paragraphs introduce the reader to the general field of study. The subsequent paragraphs then describe how an aspect of this field could be improved. The final paragraph clearly states what experimental question will be answered by the present study. The hypothesis is then stated. Next, briefly describe the approach that was taken to test the hypothesis.
I suggest the authors modify this section according to the above-mentioned points.
- - The fermentation is carried out under static conditions in the darkness for about three weeks. I would ask the authors to explain if dynamic conditions can be applied for the same purpose.
- - In the “Bacterial cellulose production” section, the authors mentioned that the content of AP was 30% or 50%, but it is unclear if these concentrations are by the weight or volume of the culture medium.
- - The Authors concluded that the introduction of apple powder into the SCOBY growing media worsened the mechanical parameters of the glycerol-modified material (mG and G_mG), and the difference was statistically significant except Young's modulus, but no discussion about the reason is not provided by the authors. I would suggest that the authors discuss how the presence of apple powder in different concentrations in the culture medium and the modification stage affects the mechanical properties of the meshes.
- - I would suggest authors provide a reference for the following paragraph and provide more information about the type of dye and dying procedure:
"To expand the range of colors, the AppBCs samples have therefore been dyed using a commercial preparation dedicated to fabric dyeing."
- -The effect of the presence of apple powder in the SCOBY culture media on the mechanical properties of the obtained bacterial cellulose has been mentioned in the conclusion section!
Author Response
We would like to thank you very much for your insightful review and suggestions for improvement. We have tried to incorporate all of them into the revised version of the manuscript and hope that we have met the expectations.
- The introduction section is too long and only covers some requirements which should be presented in this section. The introduction usually requires a short review of the literature about the research topic. Then it is best constructed as a descriptive funnel, starting with broad topics and slowly focusing on the current work. One or two paragraphs introduce the reader to the general field of study. The subsequent paragraphs then describe how an aspect of this field could be improved. The final paragraph clearly states what experimental question will be answered by the present study. The hypothesis is then stated. Next, briefly describe the approach that was taken to test the hypothesis. I suggest the authors modify this section according to the above-mentioned points.
>>> The section was re-written.
- The fermentation is carried out under static conditions in the darkness for about three weeks. I would ask the authors to explain if dynamic conditions can be applied for the same purpose.
>>> Numerous experiments of production bacterial cellulose membranes were performed and as it was observed the conditions of grown are crucial for cellulose production and strain stability. Macroscopic appearance of BC is greatly influenced by the grow conditions. In a static fermentation, a gelatinous pellicle is formed at the air-liquid interface of the culture media, whereas in an agitated fermentation, small irregular pellets are fully suspended in the culture media are produced (Cellulose (2021) 28:8229–8253, A review of bacterial cellulose: sustainable production from agricultural waste and applications in various fields. L. Urbina et al.) In addition to the shape of BC, the conditions of the process also affect the bacteria themselves. The bacterial strains cultured in static fermentation represent higher genetic stability to continuously produce BC in high yield). Whereas, the agitated fermentation often induces unfavorable conversion of bacteria into non-cellulose producing mutants what in consequence lead to decrease of the overall efficiency of the process. Of course fermentation in dynamic conditions has the advantage that it can be easily amplified to a large scale of industrial production [1], but as it was mentioned intensive agitation and aeration was found to drastically reduce cellulose synthesis [2] . ([1] Chen, G., Wu, G., Alriksson, B., Chen, L., Wang, W., Jönsson, L. J., et al. (2018). Scale-up of production of bacterial nanocellulose using submerged cultivation. J. Chem. Technol. Biotechnol. 93, 3418–3427. doi: 10.1002/jctb.5699). [2] A Krystynowicz, W Czaja, A Wiktorowska-Jezierska, M Gonçalves-MiÅ›kiewicz, M Turkiewicz, S Bielecki. Factors affecting the yield and properties of bacterial cellulose. Journal of Industrial Microbiology and Biotechnology, Volume 29, Issue 4, 1 October 2002, Pages 189–195)
In our research we did not considered agitated fermentation because it was not favorable from application point of view, for the purpose of biomaterials. BC fragments are much less useful than a whole membrane, in shape of sheet, resembling by itself a piece of leather. Therefore, static culture was the method of choice.
- In the “Bacterial cellulose production” section, the authors mentioned that the content of AP was 30% or 50%, but it is unclear if these concentrations are by the weight or volume of the culture medium.
>>> Information has been included
- The Authors concluded that the introduction of apple powder into the SCOBY growing media worsened the mechanical parameters of the glycerol-modified material (mG and G_mG), and the difference was statistically significant except Young's modulus, but no discussion about the reason is not provided by the authors. I would suggest that the authors discuss how the presence of apple powder in different concentrations in the culture medium and the modification stage affects the mechanical properties of the meshes.
>>> explanation has been included.
- I would suggest authors provide a reference for the following paragraph and provide more information about the type of dye and dying procedure:
"To expand the range of colors, the AppBCs samples have therefore been dyed using a commercial preparation dedicated to fabric dyeing."
>>> Additional information were included to the manuscript.
- The effect of the presence of apple powder in the SCOBY culture media on the mechanical properties of the obtained bacterial cellulose has been mentioned in the conclusion section!
>>> Descriptions and explanation have been added to the Result and Discussion section.
Round 2
Reviewer 1 Report
The work has been supplemented and corrected.
Author Response
We would like to thank you very much for all your corrections and very valuable comments.
Reviewer 2 Report
The manuscript is modified according to the requested modifications. Minor revisions are required as follows:
Bacterial cellulose should change to BC after its first use in the manuscript.
The following sentence should grammatically modify.
"The practical reasons behind our choice were the availability of the and its price.
Author Response

(The authors gave the same response as above.)
